# Flour Functionality, Nutritional Composition, and In Vitro Protein Digestibility of Wheat Cookies Enriched with Decolourised *Moringa oleifera* Leaf Powder

**DOI:** 10.3390/foods13111654

**Published:** 2024-05-25

**Authors:** Temitayo D. Agba, Nurat O. Yahaya-Akor, Amarjit Kaur, Moira Ledbetter, James Templeman, Jonathan D. Wilkin, Bukola A. Onarinde, Samson A. Oyeyinka

**Affiliations:** 1Centre of Excellence in Agri-Food Technologies, National Centre for Food Manufacturing, University of Lincoln, Holbeach PE12 7PT, UK; odetayotemitayo@gmail.com (T.D.A.); yahayanurat@gmail.com (N.O.Y.-A.); amarjitkaur111995@gmail.com (A.K.); bonarinde@lincoln.ac.uk (B.A.O.); 2Division of Engineering and Food Science, School of Applied Sciences, Abertay University, Dundee DD1 1HG, UK; m.ledbetter@abertay.ac.uk (M.L.); j.templeman@abertay.ac.uk (J.T.); j.wilkin@abertay.ac.uk (J.D.W.); 3Centre for Innovative Food Research (CIFR), Department of Biotechnology and Food Technology, Faculty of Science, University of Johannesburg, P.O. Box 17011, Johannesburg 2028, Gauteng, South Africa

**Keywords:** amino acid, bioactive, cookies, decolourisation, *Moringa oleifera*, wheat flour

## Abstract

This study investigated the potential of decolourised *Moringa oleifera* leaf powder (D-MOLP) in cookies to meet consumer demand for healthier food options, addressing the issue of low acceptability due to its green colour. D-MOLP and its non-decolourised counterpart (ND-MOLP) were incorporated into wheat flour to produce cookies. The results showed that neither decolourisation nor addition level (2.5 or 7.5%) significantly affected water activity or flour functionality, though slight differences in cookie colour were observed. The Moringa-enriched cookies exhibited an improved spread ratio as well as higher protein, phenolic content, antioxidant activity, and in vitro protein digestibility compared to control cookies. The detected phenolic acids included chlorogenic, ferulic, and fumaric acids, with the D-MOLP cookies showing superior nutritional properties, likely due to nutrient concentration and reduced antinutrients. Notably, glutamic acid was the major amino acid in all the cookies, but only lysine significantly increased across the cookie types. This suggests D-MOLP could be a promising alternative for food enrichment. Future research should address the consumer acceptability, volatile components, and shelf-life of D-MOLP-enriched cookies.

## 1. Introduction

The Moringa plant is a valuable source of macro- and micronutrients that could be used to address protein and energy malnutrition globally [1]. According to Asare et al. [2], all the different parts of the tree, including its flowers, leaves, and seeds are useful in folk medicine for treating different ailments. Of the various parts of the Moringa plant, the leaves are the most nutrient-dense; for example, they are an excellent source of essential amino acids and vitamins [3,4,5] and have been suggested as ingredients for food enrichment and fortification [1]. Traditionally, the fresh leaves of Moringa are consumed as a snack and integrated into salads and vegetable soups [6,7]. Other promising ways of using the leaves are through the food enrichment and fortification of commonly consumed foods like yoghurt [8], cake [9], bread [10], and biscuits or cookies [11,12,13,14,15,16,17,18,19,20].

The utilisation of *Moringa oleifera* leaf powder (MOLP) in biscuit or cookie enrichment seems to be an attractive option, especially with the growing demand for healthier foods with better functionality. Shahzad et al. [21] noted that the demand for functional foods is increasing presumably due to their perceived health benefits such as the prevention of noncommunicable diseases. Fapetu et al. [15], for example, showed that MOLP can be incorporated into wheat flour for making functional cookies with antidiabetic properties due to its inhibitory abilities against α-amylase and α-glucosidase enzymes. The Moringa plant contains phytonutrients such as carotenoids, tocopherols, and ascorbic acid [22,23], as well as bioactive compounds like quercetin and kaempferol glucoside with known pharmacological benefits [24]. This underscores the plant’s value as a versatile ingredient for diverse industrial applications. Furthermore, the remarkable ability of the Moringa plant to thrive in water-stressed environments [25] suggests its potential as a sustainable food source, aligning with Sustainable Development Goals (SDGs) 2 and 3. SDG-2 aims to ensure access to nutritious food, especially for vulnerable populations, while SDG-3 focuses on promoting health and well-being across all age groups. The incorporation of Moringa, particularly its nutrient-rich leaves in foods, could contribute significantly to achieving these goals while also supporting environmental sustainability. By reducing reliance on animal protein consumption, which is known to have a substantial carbon footprint, Moringa offers a promising solution to address both food security and climate change concerns. 

Previous research reported the use of varying levels (2–20%) of MOLP in cookies or biscuit enrichment [11,12,13,14,15,16,17,18,19,20]. Fapetu et al. [15] reported that cookies enriched with MOLP showed reduced thickness while the diameter and spread ratio increased. Furthermore, the authors found a significant increase in bioactive compounds and antioxidant properties in the MOLP-enriched cookies. According to their report, a 2.5% level of addition of MOLP was acceptable to consumers compared to other MOLP-substituted cookies. Other studies reported that MOLP can enhance the nutritional value of foods [26,27,28], improve haemoglobin levels in an adolescent [29] and confers slowly digestible starch properties and higher resistant starch content on cookies enriched with MOLP [30]. Although the use of *Moringa oleifera* has been reported to improve the nutritional value of foods, nevertheless, the appearance of enriched products is generally reported to be unacceptable [26,27,28]. The low acceptability of Moringa-enriched foods has been associated with their green colour, due to the chlorophyll content of the leaves which usually mask fortified or enriched foods [1,26,27]. For example, some authors reported that MOLP-enriched bread had an herbal flavour and generally had low acceptability with increasing levels of MOLP [31]. Ntila et al. [32] also reported low acceptability for porridge enriched with MOLP at 2–3% levels. To address this challenge and to further increase the utilisation of the leaves in food application, one practical approach would be to decolourise the leaves before its use in food enrichment. 

Earlier researchers demonstrated the use of laboratory-grade ethanol to decolourise MOLP without significant changes in nutritional value [33,34]. The decolourisation of MOLP l under optimised conditions of extraction time (10–30 min), solute-to-solvent ratio (1:20 to 1:5 *w*/*v*), and solvent concentration (50–95% *v*/*v*) using a Box–Behnken design (Response Surface Methodology) reportedly reduced the chlorophyll content of the leaves and increased protein, ash, and fibre contents [34]. Oyeyinka et al. [34] found that decolourisation under optimised conditions of solute-to-solvent ratio (1:20), solvent concentration (95%), and extraction time of 30 min can remove high levels of chlorophyll with minimal loss of certain bioactive compounds, such as phenolic compounds, into the extraction solvent. Thus, we hypothesize that the use of decolourised MOLP (D-MOLP) in cookie enrichment may produce a product with a better appearance, colour, and nutritional quality compared to cookies enriched with non-decolourised MOLP (ND-MOLP). So far, there have been no studies utilising decolourised MOLP in cookies. Hence, this study investigated the functionality of MOLP-enriched wheat flour, in terms of its physical properties, proximate composition, amino acid and phenolic acid content, antioxidant properties, in vitro protein digestibility, and microbial quality of the resulting cookies.

## 2. Materials and Methods

### 2.1. Materials

Baking items such as wheat flour (British Wheat Flour, Tesco, UK-Moisture content: 8% and Protein content: 9.9%), margarine (Stork Baking Spread, Tesco), sugar (Silver Spoon British granulated sugar), baking powder (Stockwell Eco, Tesco), salt (British cooking salt, Tesco, UK), and eggs (Large free range egg) were all purchased from Tesco Superstore Holbeach, Spalding, Lincolnshire, UK, except milk which was purchased from a local store (Powdered Peak milk, Nigeria). Moringa powder (Naturale Bio) obtained from 100% dried Moringa leaves (moisture content 7.80 ± 0.21%) was purchased from Veganok Company, India, on Amazon, UK.

### 2.2. Preparation of Decolourised Moringa Leaf Powder 

The Moringa leaves were decolourised using the optimised method of Oyeyinka et al. [34]. In brief, the optimum conditions for decolourisation of MOLP reported by these authors were a solute (MOLP)-to-solvent (ethanol) ratio of 1:20, extraction time of 30 min, and a solvent concentration of 95%. The MOLP was dissolved in the solvent, homogenized to dissolve properly, and the mixture was placed on an orbital shaker at 150 rpm for 30 min. Subsequently, the sample was centrifuged at 3200× *g* for 15 min using a centrifuge (Eppendorf, Hamburg, Germany), and the supernatant was discarded. The resulting slurry was dried at 50 °C overnight and further air-dried for 48 h to remove any residual ethanol. The dried MOLP was then packed into a thick low-density polyethylene bag for storage and use.

### 2.3. Preparation of Cookies 

For the preparation of cookies, a previously reported recipe was used with slight modifications [35]. The control cookie recipe included wheat flour (400 g), salt (4 g), sugar (200 g), milk (40 g), margarine (80 g), baking powder (8 g), and 1 egg, combined in sequence. In the enriched cookies, the wheat flour was substituted with decolourised MOLP (D-MOLP) and non-decolourised MOLP (ND-MOLP) at different ratios of 2.5% and 7.5%. This resulted in five types of cookies, including the 100% wheat flour control. The choice of 2.5% and 7.5% substitution levels was based on a preliminary study, where differences in cookie colour and appearance were not significant (*p* ≥ 0.05). Additionally, considering that higher MOLP levels provide more nutrients, these levels were chosen for the study design. The dry ingredients including wheat flour, baking powder, salt, and Moringa powder were sieved using a particle-size sieve to guarantee consistent integration and remove any foreign material. A food mixer (KitchenAid, Benton Harbor, MI, USA) was used to cream butter and sugar for 5 min at low speed until they were light and fluffy. Next, the egg and yolk were added, and the cream mixture was thoroughly mixed with the dry ingredients, followed by milk for 5 min. Then, kneading was performed for another 5 min to form dough that was not sticky and consistent, and it was refrigerated for 30 min. A circular cookie cutter was used to cut the dough into circular shapes after it had been hand-flattened onto a flat ceramic surface to a thickness of about 0.5 cm using a wooden hand roller. The cookies were baked for 20 min at 150 °C in a pre-heated oven. After baking, the cookies were carefully removed from the baking pan using gloves, placed on a tray, and covered with foil in an enclosed space.

### 2.4. Analyses on Flour Mixture with MOLP 

#### 2.4.1. Water and Oil Absorption Capacity

The ability of flour to absorb water or oil was assessed by the method described by Oladunjoye et al. [35]. For water absorption capacity (WAC), one gram of flour sample was dispersed in 10 mL of distilled water in a pre-weighed centrifuge tube. The sample was thoroughly vortexed and left to stand for 30 min at room temperature, after which the mixture was centrifuged (Eppendorf, Hamburg, Germany) at 4000 rpm for 30 min. The tube was reweighed after decanting the supernatant from it, and the WAC was expressed as the gram of water bound per gram of flour. The same procedure was repeated for oil absorption capacity (OAC), except that the water was replaced with sesame seed oil.

#### 2.4.2. Bulk Density 

The loose bulk density (LBD) and packed bulk density (PBD) of the flour samples were determined using the modified method of Kaur et al. [36] described by previous researchers [35]. Briefly, a 100 cm^3^ graduated cylinder that had previously been tarred was gently filled with the flour samples and weighed. The loose bulk density was estimated as the sample weight-to-volume ratio (g/cm^3^). To calculate the packed bulk density, the cylinder’s lid was gently tapped until no more changes in the sample level marked was observed. The packed bulk density was estimated as the sample volume-to-weight ratio (g/cm^3^).

#### 2.4.3. Foam Capacity and Foam Stability 

The foam capacity (FC) and foam stability (FS) of the flour samples were measured using the method reported by Oladunjoye et al. [35]. In a measuring cylinder, 2 g of the flour sample was dispersed in 50 mL of distilled water at room temperature. The dispersed solution was vigorously mixed and agitated to foam, and the volume was recorded after 30 s. The FC was calculated as a percentage rise in volume, whereas the FS was calculated as a percentage difference in volume from the initial amount after 1 h of whipping.

#### 2.4.4. Particle Size Distribution 

The particle size distribution of the flour samples was determined using laser diffraction analysis (Malvern Mastersizer Hydro 2000, Malvern Panalytical Ltd., Malvern, UK). Briefly, the sample was dispersed in distilled water filled in the dispersion tank while a stirrer was rotating at 2300 rpm to ensure the sample was representative due to its homogeneity. Measurements were performed in triplicate for each sample. The mean particle sizes at the 10th, 50th, and 90th percentiles of the particle size distribution curves were recorded.

#### 2.4.5. Colour 

The colour parameters (Lightness = L*, using the axis ranging between 0 for black and 100 for white, redness (+a*), greenness (−a*), yellowness (+b*), and blueness (−b*) were assessed using a bench-top chromameter (Konica Minolta, Tokyo, Japan). The total colour change (ΔE) of the flour containing MOLP was compared with the wheat control using the equation below [37].
ΔE = √〖(ΔL)〗^2 〖+(Δa)〗^2 〖+(Δb)〗^2

### 2.5. Analyses of Cookies with MOLP

#### 2.5.1. Colour and Physical Dimensions 

The colour parameters of the cookies were assessed as described in Section 2.4.5. For physical dimension measurements, including diameter, thickness, and spread ratio, six randomly chosen cookies from each sample were used as previously reported [38,39]. The diameter of the cookies was determined by taking two measurements from each cookie while rotating it 90 degrees using a ruler. The thickness was measured by taking three readings from each cookie using digital vernier callipers with a 0.01 mm accuracy, and the mean value for all six cookies were recorded. The results for thickness and diameter were expressed in millimetres (mm). The spread ratio was determined by dividing the diameter measurement by the thickness.

#### 2.5.2. Texture Analysis

The breaking strength of the cookies was measured by following the triple beam snap method (also called three-point break) using Texture Analyzer (Stable Microsystems, Godalming, UK). A cookie sample was placed on two supporting beams placed 2.5 cm apart. Another beam connected to a moving part was brought down to break the cookies at a crosshead speed at 10 mm/min and load cell of 10 kg. Care was taken to see that the point of contact was equivalent from both the supporting beams. The hardness (g/force) and brittleness were recorded, and the average values were calculated.

#### 2.5.3. Water Activity 

To record the water activity (a_w_), a calibrated water activity meter (AQUALAB 4TE, Pullman, DC, USA) was used at room temperature according to a modified method of Naknaen et al. [40]. Cookies previously milled into fine powder were put into the cup (half-filled) to prevent light from disrupting the light sensors in the meter. The readings were taken after the device made a beep sound, and this took about 5 min or less for each sample. The sample cup was cleaned after each reading.

#### 2.5.4. Proximate Composition 

Except for the moisture content, that was determined using a Halogen moisture analyser (PMB 202, Model), and the carbohydrate content, which was calculated using percentage differences [100 − (ash + fat + fibre + moisture + protein)], all the other components—the ash, protein (Micro-Kjeldahl method) and fat content (Soxhlet extraction method)—were determined using a standard method of AOAC [41].

#### 2.5.5. Total Phenolic Content and Antioxidant Capacity

The Folin–Ciocalteu reagent was used to calculate the total phenol content [42]. Using a spectrophotometer (UV–VIS spectrophotometer), absorption was measured at 765 nm and compared to a standard calibration curve made using gallic acid (Sigma Aldrich, London, UK). The findings were represented as mg Gallic acid equivalents per gram of material (mg GAE/g). 

The antioxidant capacity of the cookies was analysed using a 2,2-Diphenyl-1-Picrylhydrazyl (DPPH) assay, following the method described by Maleke and Adebo [43]. Briefly, the extract (15 µL) was treated with 285 µL of DPPH solution (working solution prepared by diluting 0.024 g of DPPH in 100 mL of methanol (MeOH) and incubating in the dark for 20 min). This was allowed to incubate at 37 °C for 15 min, and the absorbance was read at a wavelength of 570 nm on a spectrophotometer (Bibby Scientific Limited, Stone, UK).

#### 2.5.6. Phenolic Acids Quantification

The phenolic acids present in the cookies were identified by a liquid chromatograph–mass spectrometry/quadrupole time of flight (LC-MS/QToF) technique. The cookies were extracted using methanol and water in a ratio 60:40 (3 mL). The samples were diluted by a factor of 5 and filtered (0.45 µm nylon) and transferred to a HPLC vial for analysis as described by Ritvanen et al. [44]. The gradient was 0% B at 0.4 mL/min for 1 min, increasing to 100% B over 10 min, maintaining for 2 min, returning to initial conditions over 0.1 min, and re-equilibrating for 6.9 min. The MS was equipped with an electrospray ionization (ESI) source and was operated in negative ionization mode. A sample of 1 μL was injected in the column. The MS source conditions were as follows: the capillary voltage was set at 3.0 kV, the gas temperature was set at 300 °C, nitrogen was used as a drying gas at 8 L/min, the nebulizer was set at 25 psi, and a sheath gas temperature of 300 °C. The phenolic acids were quantified by external standard calibrations. The retention time, linear calibration range and R2 for each amino acid are reported in Appendix A.

#### 2.5.7. In Vitro Protein Digestibility

Protein digestibility was measured following the methods described by Awobusuyi et al. [45]. The cookies were crushed into fine powder, and 200 mg was accurately weighed into a 50 mL centrifuge tube and labelled. Then, 35 mL of 0.1 M phosphate buffer (pH of 2) and enzyme pepsin (1.5 mg/mL) were added to the tube containing the sample. The sample solution was incubated in a shaking water bath at 3 °C for 2 h to allow for digestion by the enzyme. Thereafter, 2 M NaOH (2 mL) was added to terminate the digestion, and the suspension was centrifuged at 4800 rpm at 4 °C for 20 min. The supernatant was discarded, and the residue was washed with 15 mL of a buffer solution (0.1 M phosphate, pH 7). Centrifugation was repeated as described above, and the supernatant was discarded. The residue was washed on a Whatman’s No 3-filter paper, and the undigested protein residue in the filter paper was dried in an oven (Binder, FED 260 E3.1, Tuttlingen, Germany) at 80 °C for 2 h. The protein content of the initial and final sample was determined using the Kjeldahl method, while the percentage protein digestibility was calculated as shown below.
$$Protein\;digestibility=\frac{protein\;content(undigested\;cookies-residue\;after\;digestion)}{protein\;content\;of\;undigested\;cookies}\times 100$$

#### 2.5.8. Total Amino Acid Sample Preparation and Quantification by LC-MS/QToF 

Samples were extracted as described in Section 2.5.6 and were diluted by a factor of 5 and filtered (0.45 µm nylon) and transferred to a HPLC vial for analysis. The total amino acid composition was determined by acid and alkaline hydrolysis, as described by Ritvanen et al. [44] with some modifications. The extracted proteins (100 mg) were hydrolysed in hydrochloric acid (6 M) or sodium hydroxide (5 M) at 105 °C for 24 and 18 h, respectively. The sample was centrifuged (Eppendorf centrifuge 5415 R, Hamburg, Germany) at 10,200× *g* for 2 min. Aliquots of the solution were neutralized with sodium hydroxide (5 M) and hydrochloric acid (6 M), respectively. The acid-hydrolysed samples were diluted by a factor of 50, and the alkali samples were diluted by a factor of 40. The samples were filtered (0.45 µm nylon) and transferred to an HPLC vial for analysis. The samples were further diluted as required. The amino acids in the extracted proteins were identified and quantified by LC-MS/QToF based on the method of Liu and Rochfort [46] with modifications. The system consisted of an Agilent 1290 Infinity II system including a pump, auto sampler, and column oven, and an Agilent 6546 LC/Q-ToF (Stockport, UK). The chromatographic separation was performed on a Synergi Hydro-RP column (150 × 4.6 mm, 4 µm, Phenomenex) fitted with a guard cartridge of the same stationary phase, maintained at 30 °C. Mobile phase A was water containing 0.1% formic acid, and mobile phase B was methanol containing 0.1% formic acid. The gradient was 0% B at 0.4 mL/min increasing to 100% B over 10 min, returning to the initial conditions over 0.1 min and re-equilibrating for 6.9 min. The MS was equipped with an electrospray ionization (ESI) source and was operated in a positive ionization mode. A sample of 1 μL was injected in the column. The MS source conditions were as follows: the capillary voltage was set to 3.0 kV, the gas temperature was set to 325 °C, nitrogen was used as a drying gas at 10 L/min, the nebulizer was set at 35 psi, and the sheath gas temperature was set to 325 °C. The data were collected with MassHunter Workstation LC-MS data acquisition for 6200 TOF/6500 Version11.0.221.1 (Stockport, UK). The amino acids were quantified by external standard calibrations. The retention time, linear calibration range, and R^2^ for each amino acid are reported in Appendix A.

### 2.6. Statistical Analysis 

The samples were prepared in triplicate, and the analyses were performed in triplicate unless otherwise stated. The IBM Statistical Package for Social Science (SPSS) Version 27 was used to analyse the data using one-way ANOVA, which was presented as the mean ± SD (standard deviation) of three measurements. The mean separation was by Fisher Least Significance Difference (*p* < 0.05). 

## 3. Results and Discussion

### 3.1. Functional Properties and Particle Size of Flour 

The functional properties including WAC, OAC, LBD, PBD, FC, and FS of the wheat flour were not significantly affected (*p* ≥ 0.05) by the addition of either the decolourised or non-decolourised MOLP (Table 1). The particle sizes of the wheat flour and MOLP-enriched flours were also very similar (Table 1). Although the functional properties of the wheat flour did not change significantly, there was a slight increase in its WAC and OAC, likely due to the protein content in the MOLP. In this study, the decolourised MOLP (D-MOLP) had slightly higher protein content (32.13%) compared to the non-decolourised MOLP (ND-MOLP) which had a value of 29.03%. Fapetu et al. [15] similarly reported a non-significant increase in OAC for wheat flour enriched with ND-MOLP, even up to a 10% level of addition, but a decrease in the WAC. The variation in the WAC observed in their study compared to our findings may be attributed to differences in the particle sizes of the flours, as well as variations in starch granule structure and the availability of water binding sites among the different flours used in the respective studies [26,47].

### 3.2. Water Activity and Colour of Flour

The water activity (a_w_) of food ingredients and products is indicative of shelf stability and susceptibility to microbial spoilage. It is the amount of available water in a food that supports microbial growth and participates in chemical and enzymatic reactions. The addition of MOLP generally reduced the a_w_ values of wheat flour, but the reduction was not significant (Table 1). However, MOLP addition to the wheat flour resulted in a significant change in the colour of the flour samples (Table 1). The wheat flour containing D-MOLP had much higher lightness (L*) values (86.09–89.56) compared to the wheat flour containing ND-MOLP (81.53–87.78). The decolourisation step reduced the greenness of MOLP through the removal of the chlorophyl in the leaves. This may explain the lower a* values (greenness) recorded for the wheat flour containing ND-MOLP. A previous study reported significantly lower chlorophyll content (0.56 mg/g) for D-MOLP compared to ND-MOLP (0.64 mg/g) [34]. Furthermore, the level of addition of MOLP significantly influenced the colour of the flours. Higher levels of MOLP, whether decolourised or not, increased the greenness of the wheat flour. This impacted the appearance and colour values of the cookies, as discussed in Section 3.4.

### 3.3. Geometrical Characteristics of the Cookies

The addition of MOLP generally decreased the thickness and diameter of the cookies (Table 2). However, the decrease was only significant (*p* < 0.05) at a 7.5% level of addition. The control wheat cookies recorded the highest thickness and diameter compared to all the enriched cookies. On the other hand, the spread ratio of the cookies enriched with ND-MOLP was significantly higher than that of the control and those of samples enriched with D-MOLP. This may be due to the significant decrease in thickness of the cookies. The spread ratio (7.91–10.05) of the enriched cookies in this study was higher than values previously reported (4.05–7.04) for wheat flour enriched with ND-MOLP [17,48]. A high spread ratio value is desirable and is usually associated with better cookie quality [49,50]. Differences may be observed in the spread ratio of cookies from composite flours. Watters [51] reported a decrease in spread ratio value for cookies supplemented with non-wheat flours and associated the decrease to competition by the non-wheat flours for limited free water available in the cookie dough. The competition for water is thought to result in the formation of aggregates which increases the number of hydrophilic sites in the flour [48]. During dough formation, free water is partitioned between the hydrophilic sites, thereby increasing dough viscosity [52]. Thus, the higher spread ratio in the cookies with ND-MOLP suggests that there is less competition for free water by the added ND-MOLP. 

### 3.4. Physical Characteristics of the Cookies

The addition of MOLP to wheat flour does not seem to influence the lightness (L*) values of the cookies (Table 2). However, the impact of the MOLP was significant (*p* < 0.05) on the a-values, especially for cookies enriched with ND-MOLP. In addition, a higher level of MOLP in the cookies resulted in a decrease in the a-values. A shift to a lower a-value suggests a tendency towards greenness of the sample since a positive a-value indicates redness and a negative a-value suggests greenness of the sample. MOLP contains a high level of chlorophyll which would normally mask the appearance and colour of foods enriched with it. Thus, the much lower a-value is impacted by the ND-MOLP, and this was further confirmed by the calculated total colour difference (ΔE), which was highest for cookies enriched with ND-MOLP (Table 2). This suggests that the use of D-MOLP can enhance the colour of cookies, confirming the hypothesis stated above. The sensory characteristics of the cookies, however, need further investigation since humans are the ultimate consumers. In terms of hardness and brittleness, MOLP addition generally increased hardness but decreased the brittleness of the cookies (Table 2). The hardness of cookies is influenced by the development of a gluten network [53] and water–starch–protein interactions within the ingredients used in the production of cookies [54]. The addition of MOLP to wheat flour possibly increased the viscosity of the dough, thereby increasing its hardness after baking. Previous studies also found an increase in cookie hardness following the addition of MOLP to wheat flour using instrumental methods [17,55] and subjective sensory analysis [56]. Another plausible reason for the increase in hardness is the higher level of protein in the Moringa-enriched cookies (Table 2) as explained later (Section 3.5). Proteins can absorb moisture, which can significantly affect the overall moisture balance in the dough. When proteins absorb moisture, they reduce the amount of water available for other ingredients, such as sugars and starches. This reduction in available moisture can lead to a drier dough consistency. Consequently, the cookies may have a harder texture after baking because the dough lacks the necessary moisture to maintain a soft and tender crumb. McWatters et al. [57] similarly attributed the harder texture of cookies with added cowpea to the increased protein content and its interaction with the dough during dough development. 

The Moringa-enriched cookies were very brittle compared to the control wheat cookies (Table 2). Brittle materials, even those of high strength, absorb relatively little energy before fracture. Thus, while MOLP increased the hardness of the cookies, it negatively impacted the brittleness. Optimisation may be required to obtain cookies with minimal brittleness and moderate hardness. 

### 3.5. Proximate Composition of the Cookies

The proximate composition data for the control cookies and the enriched samples are presented in Table 2. Carbohydrate (72.35–74.23%), fat (approx. 13%), and protein (8.46–9.69%) were the major components in the cookies. The ash (2.05–2.66%) and moisture content (1.52–3.24%) were generally low. The moisture content of the cookies, though significantly different (*p* < 0.05), did not follow any trend with the level of addition or decolourisation of the MOLP. All the cookie samples had low levels of moisture typical of cookies, indicating reduced likelihood of microbial growth. The moisture values obtained in this study are within the approved standard of ≤5% [58]. Moringa-enriched cookies generally showed higher protein content than the control wheat cookies. However, the level of increase at 2.5% was not significant (*p* ≥ 0.05) compared to cookies enriched at 7.5%. The cookies enriched with 7.5% D-MOLP and ND-MOLP both showed an approximately 14% increase in their protein content compared with the wheat cookies. Fapetu et al. [15] also observed a significant increase in the protein content of cookies enriched with MOLP. MOLP is a good source of protein with an appreciable amount of amino acid which is vital for health [59]. Its inclusion in food products can enhance nutritional value, addressing protein deficiencies and providing a sustainable, plant-based protein alternative, beneficial for improving overall diet quality, especially in protein-limited regions. Several other authors have reportedly used the leaves in the enrichment of various foodstuffs including cake [9], bread [10], and yoghurt [8]. 

### 3.6. Water Activity of the Cookies

All the Moringa-enriched cookies showed significantly (*p* < 0.05) lower a_w_ (0.28–0.32) compared with the control sample (0.44). Meanwhile, there was no significant (*p* ≥ 0.05) difference among the a_w_ water values for the enriched cookies. This suggests that, with or without decolourisation, the a_w_ value of the cookies is not impacted. The a_w_ values in this study are within the range of values (0.31 to 0.50) reported for cookies enriched with *Strobilanthes crispus* [60] and cookies enriched with the leaves of Mangifera indica (0.30 to 0.49) [61]. Food products with a high a_w_ value have a high risk of bacterial proliferation, destructive pathogens, and poor shelf-life [62]. Thus, the enriched cookies may have a longer shelf-life than the control wheat cookies. Studies on the shelf stability of the cookies may be required to validate this claim. 

### 3.7. Total Phenolic, Antioxidant, and Phenolic Acid Content of the Cookies

The total phenolic content (TPC) of the cookies was significantly different among the samples (Table 2). Both decolourisation and the level of addition of Moringa influenced the TPC of the cookies. Generally, the TPC increased with increasing levels of Moringa addition, which may be associated with the presence of phytochemicals, such as flavonoids and some phenolic compounds in the leaf [63]. These compounds are well-known to possess antioxidant properties with health-promoting benefits. Similarly, Moringa-enriched cookies showed higher antioxidant properties than the control wheat cookies. However, the antioxidant properties of the cookies as measured using the DPPH assay were not different among the enriched cookies, though the cookies enriched with ND-MOLP showed slightly higher values than those containing D-MOLP. Alves et al. [33] reported a decrease in the DPPH scavenging activity of MOLP after decolourisation. As earlier noted, Moringa leaves have phytochemicals with antioxidant properties. Sreelatha and Padma [64] reported varying levels of phenolic compounds in tender and matured Moringa leaves. In this study, six phenolic acids including fumaric, gallic, chlorogenic, syringic, p-coumaric, and ferulic acids were screened in the cookies using HPLC, but only three were detected in the cookies (Figure 1). Ferulic acid, which was the major phenolic acid (1263–1834 mg/100 g) in the cookies, has been suggested to be important in improving cognitive functions [65] as well as possess prebiotic activity by selectively promoting the growth of *Lactobacillus* and *Parabacteroides* in mice [66].

Chlorogenic acid was not found in the control cookies but was present in the Moringa-enriched cookies, indicating that the MOLP contributed to the chlorogenic acid in the cookies. Previous studies found that MOLP is a good source of several phenolic acids including chlorogenic acid [67,68,69,70]. These phenolic compounds are recognised for their health benefits, including anticancer properties, prevention and mitigation of oxidative stress, and reduction in cellular damage caused by free radicals [43]. The chlorogenic acid content was higher in the cookies enriched with 7.5% D-MOLP (1672 mg/100 g) and the cookies enriched with 7.5% ND-MOLP compared with those containing 2.5% levels (481 and 518 mg/100 g, respectively). However, the fumaric acid content of the cookies decreased with increasing levels of MOLP but the decrease was not significant. The control cookies showed the highest level of fumaric acid (671 mg/100 g) compared with the enriched cookies (551–631 mg/100 g).

### 3.8. In Vitro Protein Digestibility of the Cookies

The addition of MOLP significantly (*p* < 0.05) influenced the in vitro protein digestibility (IVPD) of the cookies as illustrated in Figure 2. The cookies containing D-MOLP or ND-MOLP both exhibited significantly (*p* < 0.05) higher protein digestibility (58.82–76.43%) compared to the control wheat cookies (52.54%). This improvement aligns with previous studies that reported enhanced protein digestibility of Moringa-enriched cookies [12,30,71,72]. The variation in protein digestibility within different foods can be attributed to inherent differences in food proteins and the presence of antinutrients, which affect the bioavailability of amino acids. In this study, the cookies containing ND-MOLP showed significantly (*p* < 0.05) lower IVPD than those enriched with D-MOLP. This difference is likely due to the decolourisation process, which presumably reduced the levels of antinutrients in the Moringa powder. Although the specific antinutrient levels in these cookies were not determined, previous research indicated that decolourisation of MOLP can reduce antinutrients such as tannins and phytates by approx. 60% [34]. Tannins and phytates are known to decrease the bioavailability of food nutrients by forming complexes with proteins and minerals, respectively. The IVPD results suggest that the decolourisation of MOLP could be a beneficial step in enhancing the nutritional quality of Moringa-enriched food products.

### 3.9. Amino Acid Profile of the Cookies

Moringa addition mainly improved the non-essential amino acid profile of the cookies, though lysine and threonine, which are essential amino acids, were also higher in Moringa-enriched cookies (Table 3). Glutamic acid was the major amino acid found in all the cookies. Previous studies similarly reported glutamic acid as the most abundant amino acid in Moringa-enriched cookies [73,74]. Among the essential amino acids, leucine content (456.47–493.86 mg/100 g) was the highest, while histidine was the lowest (9.98–22.48 mg/100 g). Of all the amino acids, only lysine significantly (*p* < 0.05) increased in all the cookies. Furthermore, regardless of Moringa leaf type (ND-MOLP or D-MOLP), higher levels of Moringa leaves showed higher amino acid composition. However, the Moringa cookies containing D-MOLP had slightly lower amino acid content compared with the sample enriched with ND-MOLP, though the difference was not significant (*p* ≥ 0.05). Oyeyinka et al. [34] reported a non-significant increase in histidine, serine, glycine, glutamic acid, threonine, alanine, proline lysine, tyrosine, methionine, and isoleucine when MOLP was decolourised. The improvement in non-essential amino acids is less beneficial since the body can synthesise this category of amino acids. However, considering the other benefits derivable from the cookies, such as the significant improvement in lysine, better bioactive compounds, and improved digestibility as shown in previous sections, the use of D-MOLP in food enrichment may be a welcome development in the food manufacturing sector.

## 4. Conclusions

The functional, nutritional, and in vitro protein digestibility properties of wheat–Moringa cookies have been reported in this study. The addition of Moringa flour at 2.5 and 7.5% did not change the functional properties of the wheat flour. However, the decolourisation and level of addition of Moringa leaves increase the spread ratio, protein content, antioxidant potential, and in vitro protein digestibility of cookies made from the enriched flours. Although glutamic acid was the major amino acid in the cookies, lysine was the only amino acid that increased significantly in the enriched cookies compared to the control. All the cookies except the control wheat cookies are good sources of chlorogenic, ferulic, and fumaric acids. Decolourisation did not negatively affect the bioactive compounds in the cookies including their antioxidant properties as measured by the DPPH assay. This study addresses the major challenge, of greenish discolouration imparted to foods by Moringa leaves and provides some insight into the application of decolourised Moringa leaves in cookie production. This research marks the first report on using decolourised *Moringa oleifera* leaf powder in cookie formulation. Our findings not only highlight the potential of enhancing access to nutritious food (aligned with SDG-2) but also demonstrate the feasibility of delivering a nutrient-dense staple suitable for all age groups, thereby promoting health and well-being (SDG-3). This is particularly promising given the widespread consumption of cookies across various age demographics. The decolourisation of MOLP presents a promising processing strategy applicable to food products with low consumer acceptability. Future research should focus on assessing sensory characteristics, consumer acceptability, volatile components, and the shelf-life of the cookies. Additionally, exploring environmentally friendly methods like enzymatic decolourisation is recommended for further investigation.

## Figures and Tables

**Figure 1 foods-13-01654-f001:**
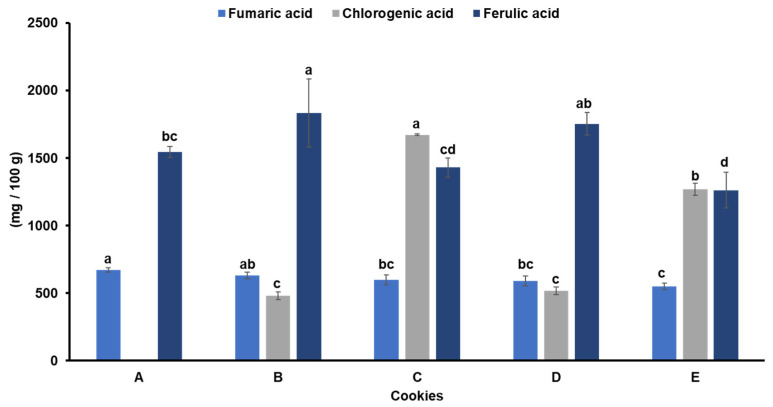
Phenolic acids of wheat–Moringa cookies. A: 100% wheat flour; B: 97.5% wheat flour + 2.5% decolourised Moringa; C: 92.5% wheat flour + 7.5% decolourised Moringa; D: 97.5% wheat flour + 2.5% non-decolourised Moringa; E: 92.5% wheat flour + 7.5% non-decolourised Moringa. Error bars indicate standard deviation (N = 3). Different letters mean significantly different (*p* < 0.05).

**Figure 2 foods-13-01654-f002:**
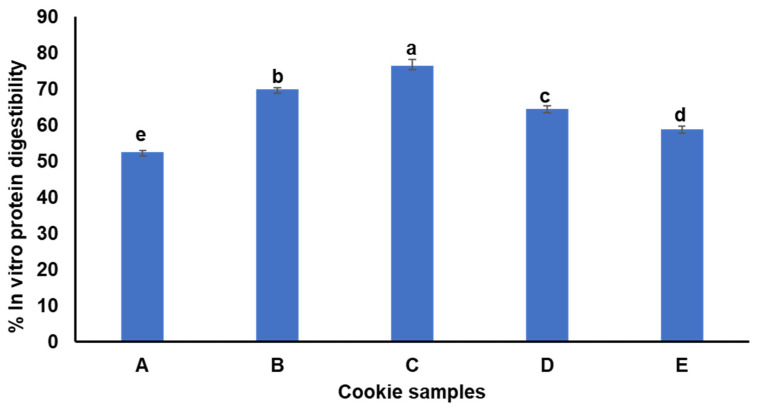
Protein digestibility of wheat–Moringa cookies. A: 100% wheat flour; B: 97.5% wheat flour + 2.5% decolourised Moringa; C: 92.5% wheat flour + 7.5% decolourised Moringa; D: 97.5% wheat flour + 2.5% non-decolourised Moringa; E: 92.5% wheat flour + 7.5% non-decolourised Moringa. Error bars indicate standard deviation (N = 3). Different letters mean significantly different (*p* < 0.05).

**Table 1 foods-13-01654-t001:** Colour, moisture, water activity, functionality, and particle size of wheat–Moringa flour.

Parameters	100% Wheat Flour	97.5% Wheat Flour + 2.5% Decolourised Moringa	92.5% Wheat Flour + 7.5% Decolourised Moringa	97.5% Wheat Flour + 2.5% Non-Decolourised Moringa	92.5% Wheat Flour + 7.5% Non-Decolourised Moringa
L*	94.04 ± 0.12 ^a^	89.56 ± 0.12 ^b^	86.09 ± 0.58 ^c^	87.78 ± 0.17 ^d^	81.53 ± 0.15 ^e^
a*	3.83 ± 0.03 ^a^	2.99 ± 0.03 ^b^	2.86 ± 0.06 ^c^	1.38 ± 0.06 ^d^	−0.04 ± 0.03 ^e^
b*	12.22 ± 0.13 ^e^	13.84 ± 0.06 ^c^	13.21 ± 0.26 ^d^	14.62 ± 0.02 ^b^	17.91 ± 0.13 ^a^
ΔE	-	4.84 ± 0.12 ^d^	8.08 ± 0.60 ^b^	7.14 ± 0.13 ^c^	14.28 ± 0.18 ^a^
a_w_	0.48 ± 0.00 ^a^	0.46 ± 0.00 ^a^	0.45 ± 0.01 ^a^	0.46 ± 0.30 ^a^	0.46 ± 0.01 ^a^
WAC (g water/g flour)	0.93 ± 0.12 ^a^	0.97 ± 0.14 ^a^	1.17 ± 0.33 ^a^	1.07 ± 0.41 ^a^	1.16 ± 0.23 ^a^
OAC (g oil/g flour)	1.27± 0.40 ^a^	1.39 ± 0.05 ^a^	1.41 ± 0.39 ^a^	1.33 ± 0.00 ^a^	1.37 ± 0.50 ^a^
LBD (g/mL)	0.50 ± 0.01 ^ab^	0.47 ± 0.00 ^b^	0.48 ± 0.00 ^b^	0.48 ± 0.00 ^ab^	0.51 ± 0.01 ^a^
PBD (g/mL)	0.67 ± 0.00 ^a^	0.66 ± 0.00 ^a^	0.66 ± 0.02 ^a^	0.67 ± 0.03 ^a^	0.69 ± 0.00 ^a^
FC (%)	28.00 ± 0.00 ^a^	26.00 ± 1.41 ^a^	26.50 ± 0.70 ^a^	26.50 ± 0.70 ^a^	26.50 ± 0.70 ^a^
FS (%)	24.00 ± 0.00 ^a^	25.50 ± 0.70 ^a^	24.50 ± 0.70 ^a^	24.00 ± 0.00 ^a^	24.50 ± 2.21 ^a^
d 0.1 (µm)	14.10 ± 1.28 ^a^	13.65 ± 0.72 ^a^	15.09 ± 0.81 ^a^	14.38 ± 0.73 ^a^	15.17 ± 0.76 ^a^
d 0.5 (µm)	79.67 ± 5.41 ^a^	80.20 ± 3.47 ^a^	84.82 ± 2.77 ^a^	85.68 ± 2.46 ^a^	83.88 ± 1.90 ^a^
d 0.9 (µm)	213.25 ± 7.36 ^a^	213.26 ± 3.37 ^a^	223.08 ± 5.88 ^a^	222.24 ± 1.75 ^a^	218.83 ± 0.80 ^a^

Values are reported as mean ± standard deviation. Mean values with different superscripts are significantly different (*p* < 0.05). WAC: Water absorption capacity; OAC: Oil absorption capacity; LBD: Loose bulk density; PBD: Packed bulk density; FC: Foaming capacity; FS: Foaming stability.

**Table 2 foods-13-01654-t002:** Physical, chemical, and physicochemical properties of cookies from wheat–Moringa flour.

Parameters	100% Wheat Flour	97.5% Wheat Flour + 2.5% Decolourised Moringa	92.5% Wheat Flour + 7.5% Decolourised Moringa	97.5% Wheat Flour + 2.5% Non-Decolourised Moringa	92.5% Wheat Flour + 7.5% Non-Decolourised Moringa
L*	47.87 ± 0.17 ^a^	49.41 ± 1.92 ^a^	47.50 ± 0.69 ^a^	48.68 ± 0.53 ^a^	46.86 ± 0.13 ^a^
a*	8.89 ± 0.50 ^a^	9.38 ± 0.46 ^a^	4.05 ± 0.23 ^b^	2.21 ± 0.61 ^c^	2.27 ± 0.11 ^c^
b*	5.55 ± 0.49 ^c^	11.63 ± 0.66 ^a^	10.13 ± 0.45 ^b^	11.46 ± 0.13 ^a^	10.70 ± 0.09 ^ab^
ΔE	-	6.41 ± 1.12 ^b^	6.70 ± 0.10 ^b^	8.97 ± 0.41 ^a^	8.45 ± 0.01 ^a^
Hardness (N)	131.97 ^b^ ± 3.97	151.31 ^ab^ ± 10.52	151.06 ^ab^ ± 2.68	151.03 ^ab^ ± 18.64	153.56 ^a^ ± 7.7
Brittleness (mm)	1.39 ± 0.35 ^a^	0.96 ± 0.05 ^b^	1.11 ± 0.14 ^ab^	0.94 ± 0.06 ^b^	0.98 ± 0.15 ^b^
Diameter (mm)	82.33 ± 2.94 ^a^	80.08± 0.98 ^ab^	78.67 ± 1.51 ^b^	81.33 ± 3.27 ^ab^	79.00 ± 2.00 ^b^
Thickness (mm)	10.35 ± 0.87 ^a^	10.12 ± 0.93 ^ab^	9.56 ± 2.02 ^abc^	8.09 ± 1.03 ^c^	8.33 ± 1.99 ^bc^
Spread ratio	7.96 ± 0.59 ^b^	7.91 ± 0.75 ^b^	8.23 ± 1.84 ^ab^	10.05 ± 1.33 ^a^	9.49 ± 2.64 ^ab^
Moisture (%)	2.05 ± 0.28 ^bc^	1.52 ± 0.11 ^c^	2.42 ± 0.11 ^b^	3.24 ± 0.37 ^a^	2.12 ± 0.25 ^bc^
Protein (%)	8.46 ± 0.03 ^b^	8.77 ± 0.20 ^b^	9.56 ± 0.05 ^a^	8.67 ± 0.03 ^b^	9.69 ± 0.16 ^a^
Fat (%)	12.40 ± 0.08 ^a^	12.86 ± 0.18 ^a^	12.78 ± 0.04 ^a^	12.38 ± 0.28 ^a^	12.45 ± 0.36 ^a^
Ash (%)	2.05 ± 0.22 ^b^	2.66 ± 0.41 ^a^	2.53 ± 0.08 ^ab^	2.08 ± 0.05 ^b^	2.36 ± 0.00 ^ab^
Carbohydrate (%)	74.23 ± 0.07 ^a^	74.23 ± 0.73 ^a^	72.35 ± 0.03 ^b^	73.28 ± 0.63 ^ab^	73.06 ± 0.76 ^ab^
a_w_	0.44 ± 0.08 ^a^	0.30 ± 0.06 ^b^	0.32 ± 0.04 ^b^	0.29 ± 0.04 ^b^	0.28 ± 0.06 ^b^
pH	7.08 ± 0.24 ^a^	6.96 ± 0.03 ^a^	6.97 ± 0.23 ^a^	6.96 ± 0.03 ^a^	6.79 ± 0.20 ^a^
Total phenolic content (mg GAE/g)	1.21 ± 0.03 ^d^	2.49 ± 0.01 ^c^	3.19 ± 0.02 ^b^	2.54 ± 0.01 ^c^	4.92 ± 0.39 ^a^
DPPH (%)	23.77 ± 1.65 ^b^	87.70 ± 1.88 ^a^	84.97 ± 5.64 ^a^	86.24 ± 0.50 ^a^	87.30 ± 0.45 ^a^

Values are reported as mean ± standard deviation. Mean values with different superscripts are significantly different (*p* < 0.05).

**Table 3 foods-13-01654-t003:** Amino acid composition of cookies from wheat–Moringa flour.

Parameters	100% Wheat Flour	97.5% Wheat Flour + 2.5% Decolourised Moringa	92.5% Wheat Flour + 7.5% Decolourised Moringa	97.5% Wheat Flour + 2.5% Non-Decolourised Moringa	92.5% Wheat Flour + 7.5% Non-Decolourised Moringa
Lysine	25.80 ± 0.33 ^c^	29.22 ± 0.63 ^b^	55.11 ± 2.19 ^a^	28.49 ± 1.07 ^b^	52.85 ± 1.19 ^a^
Histidine	11.83 ± 0.40 ^b^	11.61 ± 0.23 ^b^	22.48 ± 0.69 ^a^	9.98 ± 0.50 ^c^	22.11 ± 0.46 ^a^
Threonine	70.78 ± 1.28 ^b^	71.43 ± 0.38 ^b^	103.88 ± 4.70 ^a^	71.59 ± 0.90 ^b^	103.21 ± 2.50 ^a^
Valine	268.45 ± 16.07 ^a^	258.23 ± 3.99 ^a^	266.17 ± 21.12 ^a^	239.44 ± 22.62 ^a^	257.46 ± 11.17 ^a^
Methionine	96.54 ± 7.12 ^a^	87.69 ± 23.78 ^a^	110.92 ± 5.91 ^a^	92.79 ± 1.27 ^a^	104.32 ± 14.86 ^a^
Isoleucine	220.19 ± 7.56 ^a^	210.43 ± 1.71 ^ab^	213.86 ± 20.21 ^ab^	198.41 ± 11.34 ^b^	213.51 ± 0.66 ^ab^
Leucine	493.86 ± 12.28 ^a^	474.39 ± 10.21 ^ab^	490.56 ± 20.32 ^a^	456.47 ± 18.21 ^b^	478.07 ± 14.49 ^ab^
Phenylalanine	298.17 ± 14.40 ^a^	276.01 ± 14.26 ^a^	293.28 ± 17.15 ^a^	273.47 ± 30.62 ^a^	277.53 ± 12.38 ^a^
Arginine	47.04 ± 1.71 ^b^	50.98 ± 1.60 ^b^	142.11 ± 3.52 ^a^	43.02 ± 4.86 ^b^	143.67 ± 8.11 ^a^
Serine	20.46 ± 1.17 ^b^	20.17 ± 0.87 ^b^	32.14 ± 1.31 ^a^	23.27 ± 5.26 ^b^	33.42 ± 0.62 ^a^
Aspartic acid	72.61 ± 2.82 ^c^	75.08 ± 1.64 ^c^	102.15 ± 1.58 ^a^	86.48 ± 9.11 ^b^	100.65 ± 3.78 ^a^
Alanine	59.88 ± 2.51 ^b^	63.12 ± 0.63 ^b^	94.52 ± 4.18 ^a^	63.11 ± 1.11 ^b^	92.02 ± 2.28 ^a^
Glycine	8.90 ± 0.23 ^b^	8.89 ± 0.69 ^b^	12.29 ± 0.35 ^a^	9.86 ± 1.15 ^b^	11.61 ± 0.42 ^a^
4-Hydroxyproline	0.90 ± 0.41 ^b^	1.12 ± 0.50 ^b^	4.48 ± 0.53 ^a^	2.67 ± 2.06 ^ab^	3.35 ± 0.29 ^a^
Glutamic acid	2333.02 ± 18.28 ^a^	2268.73 ± 31.06 ^ab^	2213.10 ± 83.15 ^ab^	2111.42 ± 100.90 ^b^	2210.92 ± 39.24 ^ab^
Tyrosine	170.39 ±0.74 ^bc^	181.87 ±3.21 ^a^	164.38 ± 4.12 ^cd^	159.20 ± 5.23 ^d^	176.59 ± 10.06 ^ab^
Proline	820.93 ± 68.43 ^a^	759.43 ± 23.42 ^ab^	770.75 ± 41.67 ^a^	664.38 ± 71.68 ^b^	755.69 ± 29.76 ^ab^

Values are reported as mean ± standard deviation. Mean values with different superscripts are significantly different (*p* < 0.05).

## Data Availability

The original contributions presented in the study are included in the article/Appendix A, further inquiries can be directed to the corresponding author.

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
