# Peer review of "Flour Functionality, Nutritional Composition, and In Vitro Protein Digestibility of Wheat Cookies Enriched with Decolourised *Moringa oleifera* Leaf Powder"

_foods, 2024, doi:10.3390/foods13111654_

Round 1
Reviewer 1 Report
Comments and Suggestions for Authors
- The methodologies are described in detail, which allows us to understand the work done.
- Line 310-311 It is mentioned that the brightness (L*) of D-MOLD is lower than ND-MOLD, which is higher (this error needs to be changed).
- In the conclusions it is not mentioned whether or not the decolourisation of moringa leaves is recommended for a specific purpose (e.g. as it is mentioned that it could have health properties, which information is not sufficiently highlighted for the possible elaboration of some other by-product with D-MOLD and) since according to what is stated in this article the decolourisation would not improve the final colour of the processed biscuits compared to the non-decoloured ones. In addition, the ND-MOLD would show a better content of bioactive compounds, so the conclusion should focus on the study's importance and emphasise the benefits of adding decoloured and decoloured leaves.
Author Response
Dear Reviewer
Authors are grateful for your comments on our manuscripts. All issues raised have now been addressed in the revised manuscript. Please find below responses to your comments
We hope our responses are satisfactory.
Regards
Samson
Reviewer 1
Comments: The methodologies are described in detail, which allows us to understand the work done.
Response: We thank the reviewer for the valuable comments on our manuscript
Comments: Line 310-311 It is mentioned that the brightness (L*) of D-MOLD is lower than ND-MOLD, which is higher (this error needs to be changed).
Response: We apologise for the error. This has been changed in the revised manuscript.
Comments: - In the conclusions it is not mentioned whether or not the decolourisation of moringa leaves is recommended for a specific purpose (e.g. as it is mentioned that it could have health properties, which information is not sufficiently highlighted for the possible elaboration of some other by-product with D-MOLD and ) since according to what is stated in this article the decolourisation would not improve the final colour of the processed biscuits compared to the non-decoloured ones. In addition, the ND-MOLD would show a better content of bioactive compounds, so the conclusion should focus on the study's importance and emphasise the benefits of adding decoloured and decoloured leaves.
Response: We thank the reviewer for the observation. We have now revised the conclusion to reflect the main findings and to further highlight the health benefits. This is in addition to emphasizing the study’s importance and benefits of adding decolourised and non-decolourised moringa leaf powder.
Reviewer 2 Report
Comments and Suggestions for Authors
The demand for healthier foods with better functionality is growing. In this study, the influence of decoloration and the level of substitution of MOLP on flour properties were explored. Their effect on the geometrical characteristics, physical characteristics, composition, nutritional, and in-vitro protein digestibility properties of wheat-moringa cookies was studied. The merits and novelty of this study meet the requirements of Foods, and the study was well designed. Some questions in the manuscript need to be improved.
1. L57-61 should be included in the next paragraph.
2. L65-70 can be taken as an advantage of the usage of MOLP in cookies. The above and the following sentences can be taken as its advantages. It is better to reorganize sentences in this paragraph to make the logic more smooth.
3. According to the introduction and the content of the study, the object of this study is to investigate the potential of decolorized MOLP. So, the title should be changed accordingly to reflect the research content accurately.
4. The authors should add some basic information about the wheat flour, such as water content, and protein content.
5. The authors should briefly describe the procedure of decolorization and list the recipe of the cookie.
6. L102: Why 2.5% and 7.5% substitution were chosen without 5.0% substitution? The authors should give a brief explanation.
7. L288: The abbreviation should be used at the first appearance.
8. L293-294: The discussion should based on the significance analysis. There’s no significant difference among samples considering WAC and OAC. The following discussion is meaningless.
9. L295-296: How do you know the protein content of the MOLP, I can not find relative data from the material or Table 1.
10. Discussion should also be taken on the moisture content, fat, ash, and carbohydrate of the cookies in 3.5, especially for ones with significant changes.
11. Has it ever been reported before that the pH of the MOLP flour is much different from that of the wheat flour? If so, please add this information to the introduction. If not, I don’t think it is necessary to determine the pH of the cookie.
12. Please only kept closely related references.
Author Response
Dear Reviewer,
Authors are grateful for your comments on our manuscripts. All issues raised have now been addressed in the revised manuscript. Please find below responses to your comments
We hope our responses are satisfactory.
Regards
Samson
Reviewer 2
Comments: The demand for healthier foods with better functionality is growing. In this study, the influence of decoloration and the level of substitution of MOLP on flour properties were explored. Their effect on the geometrical characteristics, physical characteristics, composition, nutritional, and in-vitro protein digestibility properties of wheat-moringa cookies was studied. The merits and novelty of this study meet the requirements of Foods, and the study was well designed. Some questions in the manuscript need to be improved.
Response: Authors are grateful for the time and comments provided by the reviewer. All issues raised have now been addressed in the revised manuscript.
Comments: L57-61 should be included in the next paragraph.
Response: This has been addressed.
Comments: 2. L65-70 can be taken as an advantage of the usage of MOLP in cookies. The above and the following sentences can be taken as its advantages. It is better to reorganize sentences in this paragraph to make the logic more smooth.
Response: We thank the reviewer for the suggestion. We have now reorganised this section to read well, highlighting the advantages first before the challenge with utilisation.
Comments: According to the introduction and the content of the study, the object of this study is to investigate the potential of decolorized MOLP. So, the title should be changed accordingly to reflect the research content accurately.
Response: The title has been changed to reflect the study objective and the new title reads… Flour functionality, Nutritional Composition, and In-Vitro Protein Digestibility of Wheat Cookies Enriched with Decolourized Moringa oleifera Leaf Powder
Comments: The authors should add some basic information about the wheat flour, such as water content, and protein content.
Response: The moisture content: 8% and Protein content: 9.9% have now been provided
Comments: The authors should briefly describe the procedure of decolorization and list the recipe of the cookie.
Response: The procedure for decolorization including the recipe of the cookies has been provided.
Comments: The L102: Why 2.5% and 7.5% substitution were chosen without 5.0% substitution? The authors should give a brief explanation.
Response: The choice of 2.5 and 7.5% was based on a preliminary study where the difference in cookie colour and appearance was not significant (p ≥ 0.05) and considering the that the use of more MOLP will provide more nutrients, the study design decided to use these levels of MOLP addition. This has now been added to the revised manuscript.
Comments: L288: The abbreviation should be used at the first appearance.
Response: This has been corrected.
Comments: L293-294: The discussion should based on the significance analysis. There’s no significant difference among samples considering WAC and OAC. The following discussion is meaningless.
Response: We decided to discuss this section since some authors found significant difference in their study. In the current study, our result seems a bit different from the literature, we thought it was good to provide an explanation. We did mention clearly that the functional properties did not change significantly.
Comments: L295-296: How do you know the protein content of the MOLP, I cannot find relative data from the material or Table 1.
Response: We calculated the protein content of the decolourized and non-decolourized MOLP but did not report it on the table. We have added a statement next to the data as data not shown on the table since this was done to ascertain if the process affected the protein content of the leaves.
Comments: Discussion should also be taken on the moisture content, fat, ash, and carbohydrate of the cookies in 3.5, especially for ones with significant changes.
Response: The Fat, ash and carbohydrate were very similar and only protein content showed significant improvement. Since the focus was to enrich the cookies, we thought that, focusing on protein content is a better option rather than other minor components. We have added a bit of more explanation on moisture content to enrich this section.
Comments: Has it ever been reported before that the pH of the MOLP flour is much different from that of the wheat flour? If so, please add this information to the introduction. If not, I don’t think it is necessary to determine the pH of the cookie.
Response: We have deleted this section.
Please only kept closely related references.
Response: All the references have been checked and only related and relevant ones are cited and listed.
Reviewer 3 Report
Comments and Suggestions for Authors
The authors explore the potential of decolourized Moringa oleifera leaf powder for incorporation in cookie production. The study shows potential for publication, pending the fulfillment of specific requirements, as discussed below.
1) I suggest that the authors revise the title and propose a title that clearly communicates what was done and produced.
2) The aim of the work must be clear in the abstract.
3) Check the sentence from line 48 to 52.
4) Please, describe SDGs-2 and SDGs-3.
5) In the introduction, describe the presented problem and the suggested solution more thoroughly.
6) Please check line 157.
7) The results are well described; however, I suggest delving a little deeper into their discussion
8) Figure 1 does not have good quality, and the cookies show color variation between the photos. For example, Figure 1E shows darker cookies, which is not clear whether it is due to the shadow generated during the photo or if the cookie actually had this color difference.
9) Figures 2 and 3 have a difference in the size of the letters, please standardize.
10) In the conclusion, you can highlight more the novelty and impact of the themes addressed.
Comments on the Quality of English Language
The text is of good quality. It is well-written and presents information organized. However, the article should be revised to improve the fluency and clarity of the sentences.
Author Response
Dear Reviewer,
Authors are grateful for your comments on our manuscripts. All issues raised have now been addressed in the revised manuscript. Please find below responses to your comments
We hope our responses are satisfactory.
Regards
Samson
Reviewer 3
Comments: The authors explore the potential of decolourized Moringa oleifera leaf powder for incorporation in cookie production. The study shows potential for publication, pending the fulfilment of specific requirements, as discussed below.
Response: We thank the reviewer for the valuable comments to improve our manuscript. All the issues raised have now been addressed in the revised manuscript.
Comments: I suggest that the authors revise the title and propose a title that clearly communicates what was done and produced.
Response: We have now changed the title to read…. Flour functionality, Nutritional Composition, and In-Vitro Protein Digestibility of Wheat Cookies Enriched with Decolourised Moringa oleifera Leaf Powder
Comments: The aim of the work must be clear in the abstract.
Response: The aim is highlighted in red in the abstract.
Comments: Check the sentence from line 48 to 52.
Response: The sentence has been revised to read well.
Comments: Please, describe SDGs-2 and SDGs-3.
Response: Additional information has been provided on the SDGs as requested.
Comments: In the introduction, describe the presented problem and the suggested solution more thoroughly.
Response: This has been further described.
Comments: Please check line 157.
Response: Done
Comments: The results are well described; however, I suggest delving a little deeper into their discussion
Response: Further discussion have been provided where applicable throughout the manuscript.
Comments: Figure 1 does not have good quality, and the cookies show color variation between the photos. For example, Figure 1E shows darker cookies, which is not clear whether it is due to the shadow generated during the photo or if the cookie actually had this color difference.
Response: We have decided to take out the figure (Figure 1) for clarity.
Comments: Figures 2 and 3 have a difference in the size of the letters, please standardize.
Response: we have standardized but the current outlook of the figures is due to the number of parameters presented. They were both presented in Arial font, size 12.
Comments: In the conclusion, you can highlight more the novelty and impact of the themes addressed.
Response: Additional statements demonstrating the novelty and impact of the themes addressed have been provided.
Comments: The text is of good quality. It is well-written and presents information organized. However, the article should be revised to improve the fluency and clarity of the sentences.
Response: We thank the reviewer for the comments and suggestions. The revised manuscript has now been revised to improve fluence and clarity.
Round 2
Reviewer 3 Report
Comments and Suggestions for Authors
The authors addressed most of the suggested revisions. However, I believe that the discussion could be further elaborated, and the English could still be improved.
Comments on the Quality of English Language
I suggest another revision of the English to present more elaborated and clarified sentences.
Author Response
Dear Reviewer
Thank you for the additional comments on our manuscript. Additional information has been provided to improve the manuscript and the level of English has been revised again to provide more clarified and elaborated.
We hope our responses are satisfactory.
Regards
Samson